# Interrelationship between Vitamin D and Calcium in Obesity and Its Comorbid Conditions

**DOI:** 10.3390/nu14153187

**Published:** 2022-08-03

**Authors:** Iskandar Azmy Harahap, Jean-François Landrier, Joanna Suliburska

**Affiliations:** 1Department of Human Nutrition and Dietetics, Poznan University of Life Sciences, Wojska Polskiego St. 31, 60-624 Poznan, Poland; iskandar.harahap@up.poznan.pl; 2Aix Marseille University, C2VN, INRAE, INSERM, 13385 Marseille, France; jean-francois.landrier@univ-amu.fr

**Keywords:** vitamin D, calcium, obesity, physiology, comorbid

## Abstract

Obesity has been linked to vitamin D (VD) deficiency and low calcium (CAL) status. In the last decade, dietary supplementation of vitamin D and calcium (VD–CAL) have been extensively studied in animal experiments and human studies. However, the physiological mechanisms remain unknown as to whether the VD–CAL axis improves homeostasis and reduces biomarkers in regulating obesity and other metabolic diseases directly or indirectly. This review sought to investigate their connections. This topic was examined in scientific databases such as Web of Science, Scopus, and PubMed from 2011 to 2021, and 87 articles were generated for interpretation. Mechanistically, VD–CAL regulates from the organs to the blood, influencing insulin, lipids, hormone, cell, and inflammatory functions in obesity and its comorbidities, such as non-alcoholic fatty liver disease, cardiovascular disease, and type-2 diabetes mellitus. Nevertheless, previous research has not consistently shown that simultaneous VD–CAL supplementation affects weight loss or reduces fat content. This discrepancy may be influenced by population age and diversity, ethnicity, and geographical location, and also by degree of obesity and applied doses. Therefore, a larger prospective cohort and randomised trials are needed to determine the exact role of VD–CAL and their interrelationship.

## 1. Introduction

Due to inappropriate diets and nutrient intake, obesity has become more common in all age groups, such as children [1], adolescents [2], and the elderly [3]. Obesity is increasing in prevalence in Asia-Pacific [4], Europe [5], Africa [6], and America [7] and has become a worldwide epidemic [8]. Taking this into account, the member states of the United Nations have developed a global platform, the Sustainable Development Goals, addressing noncommunicable diseases as core priorities [9].

Obesity is caused by micronutrient deficiency [10], inadequate intake of vitamins, such as cobalamin (vitamin B12), ascorbic acid (vitamin C), fat-soluble vitamins, and folic acid [11], a deficiency of vitamin D (VD) [12], poor mineral status [13], and low calcium (CAL) diet [14]. The average plasma concentration of 25(OH)D is used to determine VD level. A less than 50 nmol/L of 25(OH)D serum concentration indicates VD deficiency. [15]. Furthermore, obesity is characterised by accumulation of >70% of body fat mass [16]. VD_3_, and its metabolites 25(OH)D_3_ and 1,25(OH)_2_D_3_, are deposited in adipocyte lipid droplets [17]. 25(OH)D is a fat-soluble metabolite disseminated to fat, muscle, liver, and in smaller amounts to other tissues. The stomach’s ability to absorb CAL depends on the presence of active VD. VD deficiency causes osteomalacia and rickets in adults and children, respectively, whereas moderate deficiency results in an upsurge in the risk of bone turnover and bone fractures [12].

On the other hand, CAL ion (Ca^2+^) is available in abundance in the body. Ca^2+^ is deposited in bones, mostly as CaPO_3_ (hydroxyapatite), and plays a structural role. It is required for muscle contraction and pancreatic secretion of insulin and glucagon in response to blood glucose fluctuations. Therefore, alterations in Ca^2+^ homeostasis can have a strong impact on tissues, including the heart and skeletal muscle, contributing to obesity and diabetes [18]. The adipocyte metabolism pathway is regulated by intracellular Ca^2+^. In human adipocytes, intracellular Ca^2+^ stimulates energy and fat storage through de novo lipogenesis (DNL) promotion and lipolysis prevention [19]. Hypocalcemia is characterized by CAL concentrations under 8 mg/dL or ionized CAL levels under 4.4 mg/dL [20].

Numerous experiments have established a link between vitamin D and calcium (VD–CAL) supplementation and the occurrence of metabolic diseases [21] such as chronic liver disease [22], osteoporosis [23], diabetes mellitus [24], and high blood pressure [25]. However, there are some discrepancies in the results, and the role of VD–CAL supplementation remains unclear [26]. Contradictory outcomes have been reported for VD levels [27] and CAL status [28] in obesity. Furthermore, an expert panel concluded that no confirmation supports the extraskeletal role of CAL or VD [29].

In light of current knowledge, this study assessed the interrelationship of VD–CAL in the onset of obesity and its comorbid conditions using scientific evidence from recent animal studies and human clinical trials. Besides that, this study also collected the relationship of VD–CAL in the metabolic functions of obesity, such as in the gut, bone, blood lipid and glucose levels, kidney, pancreas, liver, adipose tissue, and the immunoregulatory system. This review could aid in developing dietary recommendations for obese people and future studies into the influences of VD–CAL simulataneus supplementation in this group.

## 2. Search Strategy and Method

Scientific databases, such as Web of Science, Scopus, and PubMed, were utilised to conduct a methodical search for relevant articles by employing the following key phrases: “correlation”, “relationship”, “link”, “association”, “vitamin-D”, “calcium”, “obesity”, “rats”, “mice”, and “human clinical trial”. The search generated 87 articles published between 2011 and 2021, which investigated the relationship of VD–CAL in obesity (Figure 1). Table 1 shows the inclusion and exclusion criteria for this literature search strategy. Duplicates, review articles, protocol studies, second analysis reports, articles not related to VD–CAL, and articles not published in English were excluded. The interrelationship findings were compiled by summarising them in tables.

## 3. Physiological Association of VD–CAL

Figure 2 depicts the role of the VD–CAL axis in regulating the physiological mechanisms involved in obesity. VD–CAL status determines adipose tissue functions, body fatness, bone density, inflammatory, insulin, hormone, and cell functions.

VD exists naturally in a limited number of food sources and is endogenously formed when solar ultraviolet rays (290–315 nm) stimulate VD synthesis in the body’s outermost layer and is then primarily kept in fat in the body [30].

The available evidence indicates that VD–CAL is involved in physiological interactions; 1,25(OH)_2_D, an active VD metabolite, regulates CAL transport across the intestinal wall and binds to the VD receptor (VDR) in the intestinal epithelial cells, inducing the synthesis of CAL-binding protein CaBP-9K and activating TRPV6 and TRPV5 CAL channels. CAL enters the cell through the intestinal lumen and is transported by CAL-binding protein throughout the cell and to the interstitium via an ATP-dependent mechanism [31]. In the intestinal epithelium, 1,25(OH)_2_D modulates paracellular CAL channels in the transepithelial electrochemical gradient by regulating claudin-2 and claudin-12 [32]. In adipose tissue, it stimulates voltage-dependent and voltage-insensitive Ca^2+^ access pathways and controls Ca^2+^ release from endoplasmic reticulum stores via inositol 1,4,5-triphosphate and ryanodine receptors [16]. In bone regulation, 1,25(OH)_2_D enters cells by diffusion by binding to complex VDRs and regulates mineralised bone mass by controlling intestinal CAL absorption and supplying adequate CAL to the bone matrix [33,34]. In beta cells, activated VD metabolites increase the sensitivity of insulin receptors, glucose homeostasis transcription factors, and/or CAL regulation in peripheral insulin-target cells [30,35]. CAL in the extracellular fluid stimulates CaSR on the parathyroid cells, which raises intracellular CAL levels and suppresses PTH release. In the kidneys, PTH regulates the conversion of 25(OH)D to 1,25(OH)_2_D, which augments intestinal CAL absorption. PTH has the ability to administer circulating IL-6 and TNF-α concentrations, which promote the production of high-sensitivity C-reactive protein [32,36].

## 4. Interrelationship of VD–CAL in Obesity and Its Disease States

### 4.1. Impact of VD–CAL on Adipose Tissue

The accumulation of a large amount of fat in adipose tissue characterises obesity. The relationship between VD–CAL and obesity, both with and without supplementation, is shown in Table 2.

Adipose tissue stores most of VD, which influences CAL homeostasis and energy metabolism. A link has been shown between VD levels and CAL in adipose tissue. In obese individuals, supplementation with VD causes a significant accumulation of this vitamin in the liver and adipose tissue [38], increased leptin-to-adiponectin ratio [46], and decreased inflammation in adipose tissue [37] and hepatic steatosis by downregulating the gene expression associated with hepatic de novo lipogenesis (DNL) and oxidation of fatty acid [39]. It was also shown that administration of a high-fat diet supplemented with 1,25(OH)_2_D_3_ resulted in a reduction in macrophage infiltration in adipose tissue [40].

Besides VD intervention, supplementation of CAL in obese people was found to confer a prebiotic-like effect on the gut microbiota and decrease the stress marker expression in adipose tissue and liver inflammation [41], reduce body mass and modify glucocorticoid receptors expression and Vdr in the visceral adipose tissue [42], increase the expression of Cyp27b1/1α-hydroxylase and adipogenesis [43], decrease adipocyte hypertrophy and adipokines levels (TNF-α, IL-6, MCP-1, leptin), and rise adiponectin levels [45].

Supplementation of VD and CAL may be a useful and cost-efficient strategy for preventing and treating obesity. Rosenblum et al. [47] reported that a serving of 350 mg CAL and 100 IU VD per 240 mL glasses of orange juice resulted in significant depletion of visceral adipose tissue. Similarly, Sergeev and Song [44] observed that a high intake of VD–CAL influenced the adipose tissue Ca^2+^-mediated apoptotic pathway. The authors noted that VD_3_-CAL supplementation was more potent than either of these alone in decreasing adiposity in male C57BL/6J mice aged four weeks old.

### 4.2. Impact of VD–CAL on Body Fatness

Obesity is commonly linked to a lack of VD–CAL, which are nutrients that regulate body fat. The relationship between these nutrients and obesity is summarised in Table 3.

The prevalence of obesity may be decreased by a dose of VD or CAL. Evidence suggests that VD–CAL intake can decline body fat. The higher the concentration of 25(OH)D, the lower the body fat mass [54]. Similar to VD, a low CAL intake can negatively impact the levels of various lipid metabolic markers (glucose, triglyceride, and insulin) and increase body fat [50]. BMI and body fat levels strongly correlate with 25(OH)D concentration and the CAL-phosphorus product [56]. A high-fat diet has been revealed to promote hypermethylation and hypomethylation of Cyp24a1, resulting in glutathione deficiency and an altered VD biosynthesis pathway [51]. Additionally, there was a correlation between decreased Cyp2r1 gene expression and decreased circulating 25(OH)D concentrations [52].

However, VD–CAL levels in obese people have an unsatisfactory correlation. In obese individuals, neither BMI nor body fat percentage were significantly related to VD levels [48]. After treatment, no impact of VD supplementation was obtained on body fat, subcutaneous and visceral adipose tissue, and intrahepatic and intramyocellular lipids in this population [55]. Additionally, VD supplementation had no discernible effect on total body fat mass [57]. These results support the most recent finding that weekly VD_3_ administration at a high dose (3750 IU) had no impact on sarcopenia or adiposity indices [60].

VD–CAL have been gaining increasing interest in obesity management. These nutrients have been tested in combination and also formulated in food products. A VD-enriched *Lentinula edodes* preparation reduced total body fat accumulation and hepatic fat content in obese C57BL/6 mice [49]. Similarly, a study showed that consumption of VD-fortified yoghurt drink (containing 170 mg CAL and 12.5 μg VD_3_/250 mL) for twelve weeks led to a decrease in waist circumference, body fat mass, and truncal fat in people with type-2 diabetes aged 30–60 years old [53]. In addition, supplementation of CAL and VD_3_ caused a significant reduction in weight, BMI, waist circumference, and body fat percentage in obese women aged 18–48 years old [58]. In another study on overweight or obese males aged 18–25 years old, a 12-week supplementation of CAL and VD_3_ (600 mg of CAL and 125 IU of VD_3_) resulted in body fat loss and visceral fat loss [59].

### 4.3. Impact of VD on Bone Density

Patients and healthcare practitioners frequently overlook the importance of VD in bone health. Table 4 presents the findings of studies on the advantageous effects of VD.

Diet-induced obesity has been found to decrease the volume, number, and thickness of trabecular bone [63]. Diets containing saturated fatty acids and adequate amount of VD altered the metabolism of VD and bone changes, which suggests the critical role of dietary fat composition [65]. Supplementation of VD increased intestinal permeability and systemic lipopolysaccharide levels, resulting in stronger trabecular bone [61,62], and inhibited differentiation and maturation [64], as well as elevated autophagy flux of bone marrow-derived dendritic cells [66]. Furthermore, VD levels have been linked with bone loss [67], bone turnover [68], and areal bone mineral density [69].

### 4.4. Impact of VD–CAL on Inflammatory, Insulin, Hormone, and Cell Functions

Obesity and its associated health problems are linked to low levels of VD–CAL in the blood. VD–CAL contributes to obesity and its comorbidities by influencing inflammation, insulin sensitivity, hormone production, and cell functions (Table 5).

Apart from its vital role in CAL homeostasis, VD–CAL is involved in modulating inflammatory, insulin, hormone, and cell functions in obesity. Administration of VD was shown to reduce intestinal inflammation [77] and the level of inflammatory markers [82,83,94], and also regulate the concentrations of IL-1β, NF-Kβ, acetylcholine, brain-derived neurotropic factor [73], blood-brain barrier permeability [75], and pro-inflammatory cytokine levels [70]. However, previous studies suggest that VD–CAL supplementation does not affect inflammatory regulation in obesity. VD intervention did not cause any change in the concentrations of us-CRP and IL-6 [92], sICAM-1, hs-CRP, AGP [90], and inflammatory cytokines [89,91]. Similarly, a study showed that a CAL-rich diet had no beneficial impact on inflammation, fibrinolysis, and endothelial function [85].

Furthermore, a high-fat diet triggered insulin resistance [71], and VD deficiency also correlated with this condition in obese people [94]. VD intake influenced IRS-1/p-IRS-1 expression [80] and enhanced HOMA-IR [74]. CAL intake also improved insulin sensitivity, redox balance, and liver steatosis [83]. Nevertheless, VD supplementation had a negative impact on the relationship between 25(OH)D and insulin resistance [95] and HOMA-IR [87,96].

VD has been linked to cell regulation in obesity. The intake of this vitamin was shown to inhibit the expression of NF-κB [78] and regulate HIF1α in T cells [79]. However, VD intake had no significant effect on IFN-γ intracellular expression, NKG2D, and CD107a surface expression in NK cells, *Vdup1* and Vdr mRNA levels [76], and T cell autophagy [72]. Apart from its role in inflammatory, insulin, and cell functions, a connection has been found between VD and hormone regulation, with VD exerting a suppressing effect on PTH levels in obese individuals [84,86,88,97].

### 4.5. Potential Applications of VD–CAL in Obesity Prevention

Low levels of VD–CAL have been connected with an improved risk of obesity. Supplementation of VD–CAL can be beneficial in the treatment of obesity, as summarised in Table 6.

Obesity and its related conditions have been studied in relation to VD’s potential role as a treatment. Supplementing with VD lowered the rate of weight gain [99] and body mass [104], increased glucose homeostasis, and regulated energy expenditure in obese people [100]. Additionally, VD intake prevented histological damage of the colon [101], reduced hepatic steatosis, improved mitochondrial dysfunction in the liver, decreased stress oxidation, and triggered phase II enzymes [102].

Clinical studies, on the other hand, have found no link between 25(OH)D concentrations and hepatic enzymes, indicating a lack of interaction between VD deficiency and obesity [110] and no impact of VD on post-load glucose or other glycemic indices [112]. Supplementing with VD had no significant effect on weight, fat mass, or waist circumference [115]. Lack of VD increased the activity of neutral sphingomyelinases in obese people with cardiovascular disease, which had implications for chylomicron and low-density lipoprotein (LDL) and very-low-density lipoprotein clearance, as well as decreased postprandial inflammation and macrophage adhesion to endothelia [111].

On the other hand, in obese patients with VD deficiency, VD supplementation significantly enhanced serum 25(OH)D levels [117], reduced the size of the left atrium [113], reduced hs-CRP and TNF-α concentrations [114], and increased insulin sensitivity [116].

Furthermore, VD intake combined with physical exercise led to a rise in peak power and VD status, reduced waist-to-hip ratio [104], and improved insulin resistance and hepatic disease [98]. However, this combination did not influence inflammatory biomarkers such as CRP, TNF-α, and IL-6 [107].

The role of diets enriched with VD–CAL has also been investigated. Faghih et al. [106] reported that diet of a daily 500 kcal deficit with daily 500–600 mg CAL (1), a daily 500 kcal deficit with daily 800 mg CAL (2), a daily 500 kcal deficit with three servings of low-fat milk (3), and a daily 500 kcal deficit with three portions of CAL-fortified soy milk (4), resulted in decreased body weight and BMI in all groups of healthy, overweight, or obese premenopausal women aged 20–50 years. Valle et al. [103] showed that supplementation of salmon peptide fraction (25% of protein) and VD (15,000 IU/kg of diet) aided in maintaining metabolic syndrome via gut–liver axis by controlling hepatic and gut inflammation (increasing *Mogibacterium* and *Muribaculaceae*) in mice fed with high-fat and high-sucrose diets containing VD (25 IU/kg of diet). Sharifan et al. [108] observed a correlation between fortified dairy products containing VD_3_ (1500 IU) and anthropometric indices, glucose homeostasis, and lipid profiles in adults (30–50 years old) with abdominal obesity. Similarly, Vinet et al. [109] observed that consumption of 200 mL fruit juice supplemented with VD_3_ (4000 IU) every morning led to the weakening of microvascular dysfunction without any impact on macrovascular function by increasing 25(OH)D levels, decreasing HOMA-IR and CRP, and improving endothelium-dependent microvascular reactivity in obese adolescents aged 12–17 years old and with a BMI of 31.2–36.2 kg/m^2^.

Unfortunately, prior studies could not identify a compensatory mechanism and did not exhibit the impact of VD–CAL channels, namely TRPV5 and TRPV6, which can influence the CAL transport process. According to Veldurthy et al. [118], 1,25(OH)_2_D_3_ regulated transcellular transport by influencing the epithelial CAL channel TRPV5. TRPV5 enables the entry of apical CAL entry by inducing calbindins (calbindin-D_9k_ and calbindin-D_28k_). Moreover, calbindin-D_9k_ and TRPV6 expression are administered by 1,25(OH)_2_D_3_. TRPV6 overexpression results in hypercalciuria, hypercalcemia, and calcification of soft tissues, indicating that this channel plays a key role in the intestinal absorption of CAL.

### 4.6. Inadequate Evidence for the Effect of VD–CAL on Obesity

Although the benefits of VD–CAL supplementation have been documented, previous reports have established that these two nutrients are incompatible. The inconsistent results about the impacts of VD–CAL observed in experiments on obese populations are summarised in Table 7.

Thomas et al. [119] demonstrated insufficient evidence for the influence of high dietary CAL (CAL carbonate) on obesity-related phenotypes, including impaired weight gain and hyperphagia, in diet-induced obese 4-week-old male C57BL/6J mice. Wood et al. [120] found no effect on physical function after VD_3_ supplementation (400 or 1000 IU) for one year in postmenopausal women aged 60–70 years old with a BMI of 18–45 kg/m^2^. Salehpour et al. [121] presented inadequate data on the impact of VD_3_ intake (25 μg as cholecalciferol) on glucose homeostasis in overweight or obese women with an average age of 38 ± 8 years old and a BMI of 29.9 ± 4.2 kg/m^2^. Brzezinski et al. [122] observed an insignificant impact of 26-week VD (1200 IU) intake on body weight reduction in overweight and obese children aged 6–15 years old who had VD insufficiency (<30 ng/mL). Additionally, Jones et al. [123] showed that supplementation of dairy (~700 mg per day of CAL with 500 kcal per day) and CAL-rich dairy (~1400 mg per day of CAL with 500 kcal per day) had no effect in increasing weight loss. However, CAL-rich dairy improved plasma levels of peptide tyrosine tyrosine in men and women aged 20–60 years old and with a BMI of 27–37 kg/m^2^. Kerksick et al. [124] demonstrated that the supplementation with VD–CAL did not significantly affect the alteration of body composition in overweight postmenopausal women.

Based on the literature review, it can be concluded that the role of CAL and VD in metabolic disorders and obesity is well understood. However, the results on the impact of the deficit or the supplementation used on body weight, fat content, or biochemical parameters obtained especially in human studies are often ambiguous. It seems that many factors, including population diversity, may lead to the inconsistency of results of VD–CAL in the obese. The cited studies were conducted on a variety of populations, geographies, and races. For instance, low VD status affects the occurrence of a VD deficiency in Western European residents during the wintertime periods, South Africa, Oceania, and Asian countries (Middle East, China, Mongolia, and India) [125]. In the same vein as the global VD status, many countries have low average dietary CAL ranges between 175 and 1233 mg across half the world’s nations, such as Asia, Africa, and Latin America, and only a handful of European countries [126]. Furthermore, low VD concentrations were discovered to be a significant difference between native residents with immigrant residents [127] and multi-ethnic residents [128]. Besides that, there was an association between VD levels and elevation in a population [129]. A difference of one or two degrees in latitude enormously influenced 25(OH)D serum levels and bone density amid a low VD-sufficiency population [130]. Two different continents in the same population also influenced the CAL concentrations [131]. Therefore, these variables may predominance the inadequate evidence of VD–CAL supplementation in obese populations [120,121,122,123,124].

## 5. Limitations

Overall, the evidence from previous studies showing the importance of the VD–CAL axis in obesity suggests that investigations carried out so far, especially human studies, could not find a clear relationship between VD–CAL status in obesity [120,121,122,123,124]. There is also data showing harmful effects caused by consuming these nutrients alone or in combination [132]. However, prior studies on animal models [98,99,100,101,102,103] and humans largely [104,105,106,107,108,109,110,111,112,113,114,115,116,117] recommend the use of VD–CAL as potential therapeutic agents for the management of obesity.

Using a comprehensive literature strategy, this article has presented results collected from the studies conducted in the last ten years on the interrelationship of VD–CAL in obesity. However, the present article also has some limitations, including the lack of discussion about the side effects of VD–CAL intake, including its toxicity, which can be helpful for designing an obesity management therapy. It should be mentioned that CAL intervention studies have provided minimal evidence [133,134] supporting the application of CAL supplements in treating obesity and its comorbidities.

## 6. Conclusions and Perspectives

There is little research into the effects of the concomitant use of VD and CAL on obesity and metabolic disorders. The results of the current research, where VD and CAL were used separately and in combination, are inconclusive. Mechanistically, the VD–CAL axis affects lipid status, insulin, hormones, cells, and inflammatory functions in obesity and its comorbidities from the organs to the blood. Previous research on animals and humans has not consistently supported the hypothesis that VD–CAL supplementation can accelerate weight loss or fat loss in obesity. Many factors may affect the obtained results, including the age of subjects, degree of obesity, applied doses, population diversity, ethnicity, and geographical location. Therefore, more large-scale prospective cohort studies and randomised trials are required to fully clarify the simultaneous administration of VD–CAL as well as their interrelationship for the safe use of these nutrients in the management of the global obesity epidemic.

## Figures and Tables

**Figure 1 nutrients-14-03187-f001:**
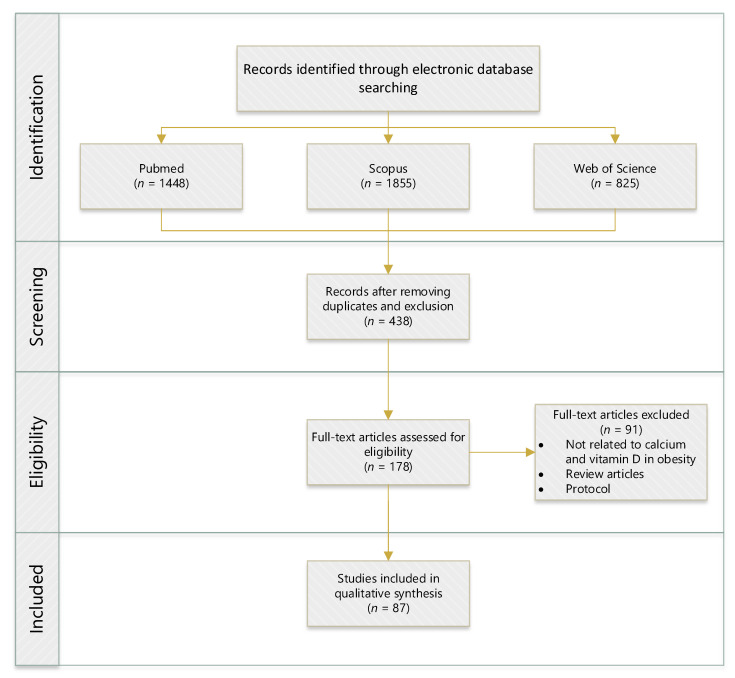
Flowchart of identifying, screening, selecting, and determining the literature search. (*n* = numbers).

**Figure 2 nutrients-14-03187-f002:**
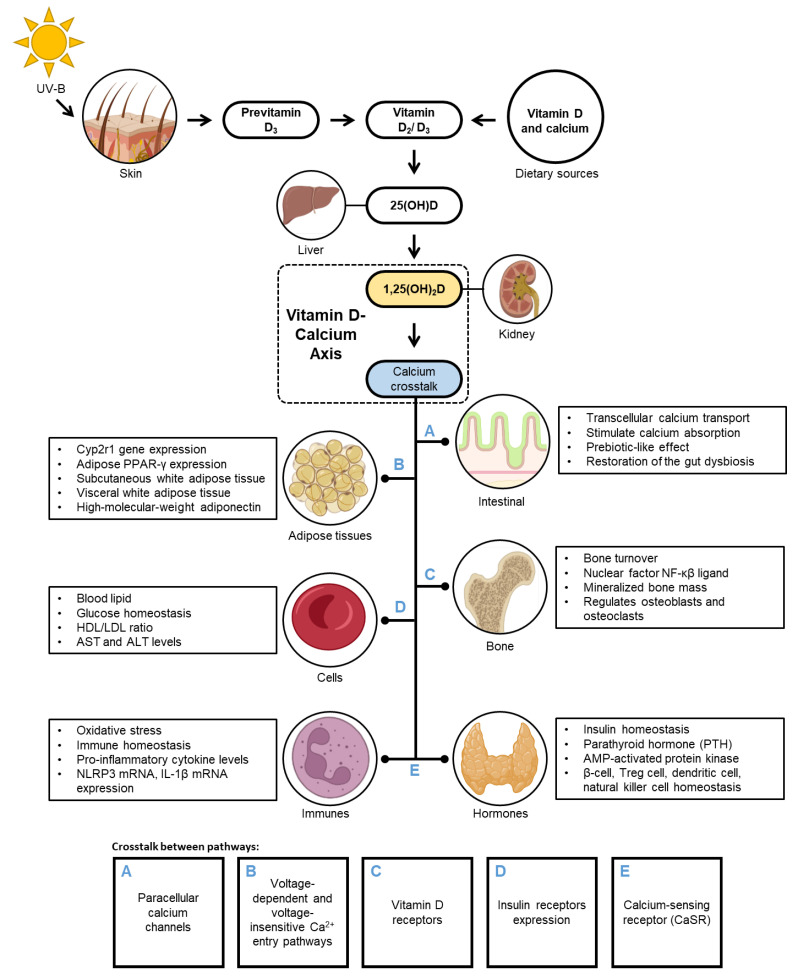
The regulation of vitamin D-calcium axis in obesity.

**Table 1 nutrients-14-03187-t001:** Search constraints and criteria.

Constraint	Inclusion	Exclusion
Type of articles	Scientific articles on animal experiments and human investigations	In vitro assessment
Relationship	Studies related to the association of either VD, CAL, or both in obese	Studies not related to the association of either VD, CAL, or both in obese
Comparator	Placebo or no comparator	—
Outcomes	There was scientific evidence of the association of either VD, CAL, or both with obesity	There was no scientific evidence of the association of either VD, CAL, or both with obesity
Study design	Evidence-based interventions, including multi-level factors, types of action, outcomes, and unintended negative consequences	Repetition articles, review journals, protocol records, case studies, second analysis reports, inadequate explanation of VD–CAL, neglect of English language, and issued earlier than 2011

VD: vitamin D; CAL: calcium; —: not to be defined.

**Table 2 nutrients-14-03187-t002:** VD–CAL in adipose tissues in animal and human studies.

Reference	Experimental Design	Diet Intervention	Comorbidities	Highlight Result
Animal study				
[37]	Male C57BL/6J wild-type mice aged eight weeks old	A high-fat diet with 500 IU/kg diet followed by a 10,000 IU/kg diet of VD_3_	Obesity	Gut and adipose tissue cross-talked in VDRs signalling associated with lipid homeostasis.
[38]	C57BL/6 mice aged six weeks old	High-fat-diet with 25,000 IU of VD_3_	Obesity	↑ VD_3_ concentrations in liver and adipose tissue
[39]	Male C57BL/6J mice aged six weeks old	High-fat and high-sucrose with 15,000 IU/kg of VD	Obesity	↓ chemokine mRNA concentrations in adipose tissue; ↓ lipid droplets and triglyceride accumulation in the liver; ↓ hepatic DNL and fatty acid oxidation expression.
[40]	New-born Wistar rats	1 μg/kg of 1,25(OH)_2_D_3_	Obesity	↓ macrophages in pancreas and adipose tissue inflitration; ↑ Tgf-β1 in liver tissue expression; ↑ Treg cells and insulin-targeted tissues inflitration.
[41]	Obese C57BL/6J mice aged five weeks old	A high-fat diet with 4 g/kg of CAL	Obesity	↓ endotoxin concentrations; ↑ angiopoietin-like 4 expression; ↓ hepatic lipid; ↑ adipose tissue expression and inflammation.
[42]	Obese Wistar rats	10 g CaCO_3_/kg	Obesity	↓ VAT fat acid synthase, ↓ leptinemia; and ↓ Vdr.
[43]	Early weaning Wistar rats	10 g CaCO_3_/kg	Obesity	↑ calbidin, VDR and prevented adipose tissue dysfunction
[44]	Four-week-old C57BL/6J male mice	High CAL, high VD	Obesity	↓ weight gain (body and fat), ↑ adiposity and ↑ plasma glucose, insulin, adiponectin, calcifediol, calcitriol, PTH.
[45]	Obese male Wistar rats with CAL deficiency	1000 mg of CAL carbonate per 100 g high-fat diet	Obesity	↓ body weight, ↓ adiposity, ↓ glucose, ↓ insulin, ↓ HOMA-IR, ↓ TNF-α, IL-6, MCP-1, Leptin, ↓ hepatic lipid, ↓ hepatic macrophage, ↓ adipocyte hypertrophy, and ↑ adiponectin level
Human study				
[46]	Obese adolescent patients aged 14.1 + 2.8 years old	4000 IU of VD_3_	Obesity	↑ serum 25(OH)D; ↑ fasting insulin; ↑ HOMA-IR; ↑ leptin-to-adiponectin ratio; No changes in inflammatory signs.
[47]	Men and women with overweight and obese aged 18–65 years old	350 mg of CAL and 100 IU of VD_3_	Obesity	Significant correlation of VD–CAL on visceral adipose tissue.

VD: vitamin D; CAL: calcium; ↑: increase; ↓: decrease.

**Table 3 nutrients-14-03187-t003:** VD–CAL in body fatness.

Reference	Experimental Design	Diet Intervention	Comorbidities	Highlight Result
Animal Study				
[48]	Obese male C57BL/6J mice aged twenty four weeks old	162 IU, 1282 IU and 5169 IU of VD_3_/kg	Obesity	↓ serum calcitriol; ↑ serum parathyroid hormone.
[49]	Male C57BL/6 mice aged ten weeks old	A high-fat diet with VD-enriched mushrooms extract	Obesity	↓ total body fat; ↓ hepatic fat content; ↑ immunomodulatory effect (CD4/CD8 lymphocyte ratio).
[50]	Wistar and genetically predisposed obese IIMb/b rats	A low dose of 0.2% CaCO_3_.	Obesity	↑ body fat, ↑ liver weight, ↑ perigonadal and retroperitoneal fat in low CAL intake; ↓ body ashes and ↓ total skeleton bone mineral in low CAL intake.
[51]	Male C57BL/6J mice aged five weeks old	A high-fat diet	Obesity	↓ CYP2R1 and CYP27A1, CYP27B1, VDR; ↑ CYP24A1 (24-hydroxylase); ↑ DNA methylation, Dnmt activity, and 5-methylcytosine; ↓ Tet activity and 5-hydroxymethylcytosine.
[52]	C57BL/6 mice	A high-fat diet	Obesity	↓ Cyp2r1 mRNA; Positive relationship between Cyp2r1 mRNA levels with 25(OH)D levels and cholecalciferol ratio.
Human Study				
[53]	Type 2 diabetes patients aged 30–60 years old	Fortified yogurt drink by 170 mg CAL with 12.5 μg VD_3_/250 mL twice a day	Diabetic type 2	↓ waist circumference, ↓ fat mass, ↓ truncal fat, and ↓ visceral adipose tissue
[54]	Healthy overweight or obese women with an average age of 38 ± 8.1 years old	Cholecalciferol 25 μg	Obesity	↓ serum iPTH levels; ↓ fat mass; No alterations in body weight and waist circumference.
[55]	Healthy adults aged 18–50 years old with BMI > 30 kg/m^2^	7000 IU of VD each day	Obesity	No alterations in body fat, intramyocellular lipids, VAT, intrahepatic, subcutaneous, HOMA, blood pressure, plasma lipids, and hsCRP.
[56]	Males and females aged 35–51 years old with BMI ≥30 kg/m^2^	No intervention of VD	Obesity	Negative correlation among 25(OH)D levels, BMI, and percentage body fat.
[57]	Thin and obese women aged 57–90 years old with VD insufficiency (50 nmol/L)	Daily doses of VD_3_ of 400, 800, 1600, 2400, 3200, 4000, and 4800 IU	Obesity	No significantly different in total body fat mass.
[58]	Obese women aged 18–48 years old	50,000 IU of cholecalciferol/week, 1200 mg/dL CAL/day, and cholecalciferol plus CAL	Obesity	↓ body fat percentage, ↓ fasting blood glucose, ↓ PTH, ↓ cholesterol, and ↓ triglycerides in cholecalciferol plus CAL intake.
[59]	Overweight or obese male aged 18–25 years old	CAL carbonate (600 mg) and VD_3_ (125 IU)	Obesity	↓ fat mass loss; ↓ visceral fat mass and ↓ visceral fat area.
[60]	Overweight and obese males and females aged greater than 65 years old	600 IU and 3750 IU of VD	Obesity	No significant differences in muscle and visceral adiposity

VD: vitamin D; CAL: calcium; ↑: increase; ↓: decrease.

**Table 4 nutrients-14-03187-t004:** VD on bone density.

Reference	Experimental Design	Diet Intervention	Comorbidities	Highlight Result
Animal study				
[61]	C57BL/6J male and female offspring	High and low VD	Obesity	Positive correlation between *Bacteroides* and LPS, trabecular femur peak load, vertebral trabecular separation, trabecular number, and bone volume fraction in both
[62]	Three-week-old C57BL/6J mice	High or low VD; high fat and sucrose	Obesity	↓ lower intestinal permeability; ↑ trabecular number and ↓ trabecular separation; No effects on IL-6 and TNF-α serum concentrations.
[63]	Twenty-one-week-old male obese Sprague Dawley rats underwent Roux-en-Y gastric bypass	High-fat diet	Obesity	↓ serum bicarbonate, calcium, 25-hydroxyvitamin D, insulin, and leptin levels; ↑ serum osteocalcin; ↓ P1NP levels; ↓ the volume, number, and thickness of trabecular bone.
[64]	Male C57BL/6N obese mice aged five weeks old	High-fat diet	Obesity	↓ Il12b mRNA levels in stimulated bone marrow-derived dendritic cells
[65]	Female C57BL/6J mice aged eight months old	Saturated fatty acids and VD 1000 IU/kg diet	Obesity	↓ hepatic Cyp2r1 and renal Cyp24a1 mRNA expression
[66]	C57BL/6 males aged ten weeks old	High-fat diet, VD 1000 and 10,000 IU/kg diet	Obesity	↓ phenotypes related to dendritic cells function expression; ↓ production of IL-12p70 by bone marrow-derived dendritic cells; ↑ LC3 Ⅱ/Ⅰ and VPS34 protein concentrations; ↓ p62 expression; ↓ Vdr mRNA levels.
Human study				
[67]	Healthy overweight and obese women	VD_3_ doses (100,000 IU; 3420 IU; and 3420 IU)	Obesity	↓ BMD; ↑ bone turnover markers; normal parathyroid hormone concentrations; No association between weight loss and changes in BMD; Positive association between low levels of bone loss and high levels of 25(OH)D.
[68]	Healthy males and females aged 18–50 years old	7000 IU cholecalciferol	Obesity	↑ plasma 25OHD, ↓ PTH, ↓ CTX; Inverse correlation between alterations in plasma 25(OH)D, bone-specific alkaline phosphatase, and CTX; Inverse correlation between alterations in CTX and in spine BMD; Association between ↑ levels of 25OHD with ↓ PTH, ↓ bone turnover, and ↑ BMD at the forearm.
[69]	Overweight and obese children aged 8–11 years old	No intervention of VD	Obesity	A link between 25(OH)D and areal BMD.

VD: vitamin D; CAL: calcium; ↑: increase; ↓: decrease.

**Table 5 nutrients-14-03187-t005:** VD–CAL in inflammatory, insulin, hormone, and cell functions.

Reference	Experimental Design	Diet Intervention	Comorbidities	Highlight Result
Animal study				
[70]	Male BALB/C mice	A high-fat diet	Obesity	↑ cytokine levels and IL-1b mRNA expression; Negative correlation between VD levels, inflammatory cells, IL-1b and IL-17 levels.
[71]	Male Wistar rats	A high-fat diet	Obesity	↓ intracellular CAL transient rates, ↓ rates and amplitude of salivary acinar cells
[72]	Males C57BL/6 mice aged five weeks old	Diet-induced obesity	Obesity	↓ autophagy of T cells, ↑ dysregulation of T cell homeostasis
[73]	Male Wistar rats	VD 500 IU/kg diet	Obesity	↓ food intake and weight gain; ↓ hippocampus acetylcholine concentrations; ↑ hippocampus IL-6 concentrations.
[74]	Male Wistar rats	VD 500 IU/kg diet	Obesity	↓ weight and food intake; ↓ insulin resistance; No influence of VD on degenerated neurons and TNF-α concentration.
[75]	Hippocampus male Wistar rats	VD 500 IU/kg diet	Obesity	↓ body weight, NF-κB concentrations, blood–brain barrier permeability; ↑ increased brain-derived neurotrophic factor concentrations.
[76]	Male C57BL/6 mice aged five weeks old	VD 10,000 IU/kg diet	Obesity	↓ natural killer cells
[77]	C57BL/6J mice	A high-fat/high-sugar diet with low amounts of VD	Obesity	↓ steatosis and fibrosis; ↓ inflammatory and pro-fibrotic genes; ↓ intestinal inflammation.
[78]	Male C57BL/6J mice aged four weeks old	A high fat-high sugar with VD	Obesity	↓ NF-κB activation, TNF-alpha level, SCAP/SREBP lipogenic pathway activation, CML protein adducts level, and RAGE expression.
[79]	Male C57BL/6N mice aged ten weeks old	A high-fat diet with 1000 or 10,000 IU of VD/kg diet	Obesity	↑ CD4 + IL-17 + T cells; ↑ CD4 + CD25 + Foxp3 + T cells; ↓ phospho-p70S6K/total-p70S6K ratio; ↑ phospho-AKT/total-AKT ratio; ↓ Hif1α mRNA levels.
[80]	Obesity-related diabetes rats	VD	Obesity	↓ parathormone and adipocytokines
[81]	Obese male Wistar rats	500 IU/kg VD	Obesity	↑ superoxide dismutase activity; ↓ catalase activity; ↓ TNF-α concentration in heart tissue.
[82]	Obese male Wistar rats	500 IU/kg VD	Obesity	↓ glutathione peroxidase activity; ↓ hepatic tumour necrosis factor concentrations; ↑ superoxide dismutase activity; No effects on glutathione peroxidase or catalase activity.
[83]	Female Wistar rats	10 g CaCO_3_/kg diet	Obesity	↑ fatty acid synthase, ↑ steatosis, and ↓ protein kinase B (Akt)
Human study				
[84]	Men and women aged 30–80 years old	1000, 2000, and 4000 IU VD_3_ per day	Obesity	↓ PTH levels with 1000 IU per day; ↑ PTH levels with 2000 to 4000 IU per day.
[85]	Patients aged 20–60 years old	A high-CAL diet (1200–1300 mg/d) and low-CAL diet (<500 mg/d)	Obesity	↓ inflammation markers, ↓ fibrinolysis, and ↓ endothelial dysfunction.
[86]	Males and females aged 18–59 years old	50,000 IU of VD per week	Obesity	↓ PTH, MCP-1, IL-1β and TLR-4
[87]	Obese and overweight women aged 20–40 years old	50,000 IU VD per week	Obesity	No alterations in C-reactive protein, insulin, insulin resistance, and waist to hip ratio.
[88]	Overweight and obese adults (aged 32 ± 8.5 years old)	Cholecalciferol doses 100,000 IU followed by 4000 IU per day	Obesity	No influence in iPTH and cFGF-23; Inverse correlations between serum iPTH and cFGF-23.
[89]	Obese children and adolescents aged 12.89 ± 1.63 years old	VD 120,000 IU/month and 12,000 IU/month	Obesity	No significant difference in insulin resistance, sensitivity, inflammatory cytokines, and pulse wave velocity
[90]	Pregnant women	400 IU/d to 150,000 IU per three months	Obesity	Positive association: BMI and 9 inflammatory biomarkers; Positive association: BMI and sICAM-1, hs-CRP, and AGP; Inverse association: 25(OH)D and sICAM-1, hs-CRP, and AGP; No impact of VD intake on inflammatory biomarkers.
[91]	Overweight and obese youth aged 11–17.99 years old	150,000 IU ergocalciferol per 3 months	Obesity	No alteration in inflammatory markers
[92]	Obese patients aged between 10 and 65 years.	No intervention of VD	Obesity	↓ blood levels of VD; ↑ US-CRP, IL-6, HOMA
[93]	Obese patients aged > 18 years old	No intervention of VD	Obesity	↓ metabolic status, ↑ liver enzymes, ↑ inflammatory markers
[94]	Healthy children aged 9–13 years old	No intervention of VD	Obesity	↑ VD insufficiency occurrence in children with IR
[95]	Obese adolescents aged 12.7 ± 1.3 years old	No intervention of VD	Obesity	Negative association: 25(OH)D and HOMA-IR with alanine aminotransferase
[96]	Obese and overweight postmenopausal women without diabetes	No intervention of VD	Obesity	Inverse association: 25(OH)D and fasting and 2-h insulin, HOMA-IR, and PTH
[97]	Overweight and obese children aged 2-18-year-old	No intervention of vitamin D	Obesity	↓ serum 25(OH) D level; ↓ PTH

VD: vitamin D; CAL: calcium; ↑: increase; ↓: decrease.

**Table 6 nutrients-14-03187-t006:** The evidence of VD–CAL supplementation in obesity.

Reference	Experimental Design	Diet Intervention	Comorbidities	Highlight Result
Animal study				
[98]	Male C57BL/6J mice aged ten weeks old	High-fat/sucrose (HFS), physical exercise, and VD	Obesity	↑ insulin sensitivity and hepatic steatosis in combining physical exercise and VD; ↓ hepatic de novo lipogenesis in combining physical exercise and VD.
[99]	Adult Wistar strain albino female rats	Monosodium glutamate and calcitriol	Obesity	↓ body weight, food, and water intake
[100]	Male C57BL/6J mice aged six weeks old	VD_3_ 15,000 IU/kg diet	Obesity	↑ lipid oxidation; ↑ upregulation of fatty acid oxidation and mitochondrial metabolism genes; ↑ expenditure on energy.
[101]	Male C57BL/6N mice aged five weeks old	VD 1000 and 10,000 IU/kg diet	Obesity	↑ colonic Cldn1 and Cyp27b1 mRNA levels; Negative relationship: 25(OH)D levels and histology score.
[102]	Male Sprague Dawley rats aged six weeks old with high-fat-diet-induced NAFLD	50 mg/kg coral CAL and 50 mg/kg coral CAL hydride	Obesity	↓ body weight gain, ↑ hepatic mitochondrial dysfunction, ↓ oxidative stress, and activated phase II enzymes in coral CAL hydride treatment.
[103]	Low-density lipoprotein receptor (LDLr) mice	Salmon peptide fraction and VD_3_ (15,000 IU/kg of diet)	Obesity	↑ metabolic syndrome via a gut-liver axis ↑ *Mogibacterium* and *Muribaculaceae*)
Human study				
[104]	Overweight and obese women	400 IU of VD per day	Obesity	An association weight loss and ↑ 25(OH)D levels.
[105]	Overweight and obese adults aged 26.1 ± 4.7 years old with a BMI 31.3 ± 3.2 kg/m^2^	4000 IU of VD per day	Obesity	↑ 25(OH) D; ↓ PTH; Inverse relationship between 25(OH)D and waist-to-hip ratio.
[106]	Overweight or obese premenopausal women aged 20–50 years old	500–600 mg/day CAL, 800 mg/day CAL, low-fat milk (1.5%), and soy milk fortified CAL 1200 and 1300 mg/day.	Obesity	↓ weight and BMI in all interventions
[107]	Overweight and obese patients	4000 IU of VD per day with exercise training	Obesity	A significant correlation between percent body fat and CRP, and between serum 25OHD and CRP; A significant reduction in unstimulated TNFα production both groups.
[108]	Adults aged 30–50 years old with abdominal obesity	Fortified low-fat yogurt (1500 IU VD_3_ per 150 g/d); fortified low-fat milk (1500 IU VD_3_ per 200 g/d)	Obesity	↑ 25(OH)D serum levels; ↓ weight to hip ratio; ↑ triglyceride and HDL-C ↓ fasting serum insulin ↑ HOMA-IR and quantitative insulin sensitivity
[109]	Obese adolescents aged 12–17 years old and with BMI 31.2–36.2 kg/m^2^	Fruit juice with VD 4000 IU per day	Obesity	↑ total and free 25(OH)D; ↓ HOMA-IR and CRP; ↑ endothelium-dependent microvascular.
[110]	Overweight, obese patients aged 18–60 years old	4000 IU cholecalciferol/day	NAFLD and NASH with VD-deficient	No impact between VD and hepatic enzymes
[111]	Obese, pre-menopausal adult women	75,000 IU cholecalciferol	Cardiovascular with VD-deficient	↑ activity of neutral sphingomyelinases, ↓ chylomicrons, ↓ LDL and VLDL, ↓ postprandial inflammation and endothelial macrophage adhesion
[112]	Overweight and obese African Americans aged 57.0 ± 10.4 years old	VD_3_ (4000 IU/day)	Prediabetes with VD-deficient	↑ 25OHD level; No impact of VD on post-load glucose or other glycemic measures.
[113]	Obese male veterans aged 35–85 years old	50,000 IU ergocalciferol/week	Prediabetes with 25(OH)D level 5.0–29 ng/mL	↑ left atrial volume ergocalciferol, No significant difference in blood pressure and other diastolic parameters
[114]	Diabetic patients aged 66.3 ± 4.4 years old	30,000 IU cholecalciferol/week	Type 2 diabetes with VD deficient or insufficient	↓ hs-CRP and TNF-α concentrations
[115]	Male and female 30–60-year-olds with obesity and type 2 diabetes	Stage 1: 6000 IU of VD/day; Stage 2: 3000 IU of VD/day; Stage 3: 2200 IU of VD/day	Obesity and type 2 diabetes	No associations between VD supplementation with weight, fat mass, or waist circumference
[116]	Obese males and females aged 18–70 years old	Cholecalciferol 25,000 IU/week	Obesity with VD deficient	↑ insulin sensitivity
[117]	Obese children aged 2–14 years old	50,000 IU cholecalciferol/week	Obesity with VD deficiency	↑ VD deficiency status after consuming the dose of cholecalciferol

VD: vitamin D; CAL: calcium; ↑: increase; ↓: decrease.

**Table 7 nutrients-14-03187-t007:** The inadequate evidence of VD–CAL status in obesity.

Reference	Experimental Design	Diet Intervention	Comorbidities	Highlight Result
Animal study				
[119]	Obese male C57BL/6J mice aged four weeks old	A high-fat diet, high-CAL, and high-CAL + non-fat dry milk	Obesity	↓ body weight and adiposity; ↑ adiposity in high-CAL diet; ↑ feed efficiency and hyperphagia develop obesity in high-Ca mice; A strong correlation between mRNA markers of macrophages.
Human study				
[120]	Healthy post-menopausal women aged 60–70 years old	400 and 1000 IU of VD_3_	Obesity	No improvement in physical function (grip strength or falls)
[121]	Overweight and obese women aged 38 + 8 years old with a BMI 29.9 + 4.2 kg/m^2^	25 μg of VD_3_	Obesity	↓ fasting blood glucose concentrations; ↑ 25(OH)D concentrations; ↓ percentage of HbA1c; Significant association: HbA1c and 25(OH)D concentrations.
[122]	Overweight and obese children aged 6–15 years old	1200 IU of VD_3_	Obesity	↓ BMI not significant; ↓ fat mass not significant; No impact on body weight reduction
[123]	Men and women aged 20–60 years old with a BMI 27–37 kg/m^2^	Dairy with 700 mg/day CAL, and 1400 mg/day CAL	Obesity	↑ weight loss and ↑ plasma levels of peptide tyrosine tyrosine (PYY) in high CAL fortification.
[124]	Overweight females with postmenopausal aged around 51.3 years old	Daily 800 mg of CAL citrate and malate + 400 IU of VD	Overweight with postmenopausal	Negligible impact on body composition

VD: vitamin D; CAL: calcium; ↑: increase; ↓: decrease.

## Data Availability

Data sharing not applicable.

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
