# Peer review of "Interrelationship between Vitamin D and Calcium in Obesity and Its Comorbid Conditions"

_nutrients, 2022, doi:10.3390/nu14153187_

Round 1

Reviewer 1 Report

Nutrients:

Interrelationship between vitamin D and calcium in obesity 2 and its comorbid conditions

General comments:

Several observational studies have investigated the relationship between vitamin D deficiency and risk of central obesity. According to the literature, a systematic search was carried out of all published articles, in five electronic databases, revealed that the highest versus the lowest serum vitamin D level was significantly associated with a 23% decreased odds of abdominal obesity (Hajhashemy, Z., Shahdadian, F., Ziaei, R., & Saneei, P. (2021). Serum vitamin D levels in relation to abdominal obesity: A systematic review and dose-response meta-analysis of epidemiologic studies. Obesity reviews : an official journal of the International Association for the Study of Obesity22(2), e13134. https://doi.org/10.1111/obr.13134). Also, many systematic reviews confirmed that.

Why the authors wanted to discuss again through a review the Vitamin D and Calcium topic ?

Specific comments:

1.      Abstract:

The abstract is very poor and doesn’t look like describing the methods of search nor the main results or the main findings. It needs to be reformulated

2.      Line 76: Do we really need to know about facts concerning vitamin D and Calcium?

The physiological relationship is already known. Why the authors need to repeat this information? I suggest to remove these paragraphs and to replace them by visual tool that minimize length and time consumed when reading this manuscript. Also, I suggest to include this reference to talk about these mechanisms: El-Hajj Fuleihan G. (2012). Can the sunshine vitamin melt the fat?. Metabolism: clinical and experimental61(5), 603–610. https://doi.org/10.1016/j.metabol.2011.12.006.

3.      Starting Table 3, I did some search of pubmed and found many articles are missing. There are many articles that included studies on obese people or people with overweight and they were not included in this manuscript. For instance: Jabbour, J., Rahme, M., Mahfoud, Z. R., & El-Hajj Fuleihan, G. (2022). Effect of high dose vitamin D supplementation on indices of sarcopenia and obesity assessed by DXA among older adults: A randomized controlled trial. Endocrine76(1), 162–171. https://doi.org/10.1007/s12020-021-02951-3. I suggest for authors to recheck again the literature to include all studies.

Discussion: I didn’t notice any paragraph that discuss the findings obtained in the review with other data. Authors should elaborate this paragraph to allow comparison between countries, ethnicities,.,,also to 

Author Response

Dear Editor and Reviewers, We are very grateful for your comments on our manuscript. We have revised the manuscript in accordance with your advice, and carefully proof-read it to minimize any errors. Reviewer 1 1/ Several observational studies have investigated the relationship between vitamin D deficiency and risk of central obesity. According to the literature, a systematic search was carried out of all published articles, in five electronic databases, revealed that the highest versus the lowest serum vitamin D level was significantly associated with a 23% decreased odds of abdominal obesity (Hajhashemy, Z., Shahdadian, F., Ziaei, R., & Saneei, P. (2021). Serum vitamin D levels in relation to abdominal obesity: A systematic review and dose-response meta-analysis of epidemiologic studies. Obesity reviews : an official journal of the International Association for the Study of Obesity, 22(2), e13134. https://doi.org/10.1111/obr.13134). Also, many systematic reviews confirmed that. Why the authors wanted to discuss again through a review the Vitamin D and Calcium topic ? Response: In this review we would like to focus on calcium and vitamin D relationship and also on the influence of vitamin D and calcium taken simultaneous on obesity. In the Introduction we added the justification of this review. 2/ The abstract is very poor and doesn’t look like describing the methods of search nor the main results or the main findings. It needs to be reformulated Response: The abstract has been rewritten. 3/ Line 76: Do we really need to know about facts concerning vitamin D and Calcium? The physiological relationship is already known. Why the authors need to repeat this information? I suggest to remove these paragraphs and to replace them by visual tool that minimize length and time consumed when reading this manuscript. Also, I suggest to include this reference to talk about these mechanisms: El-Hajj Fuleihan G. (2012). Can the sunshine vitamin melt the fat?. Metabolism: clinical and experimental, 61(5), 603–610. https://doi.org/10.1016/j.metabol.2011.12.006. Response: This part has been changed and the reference has been included. 4/ Starting Table 3, I did some search of pubmed and found many articles are missing. There are many articles that included studies on obese people or people with overweight and they were not included in this manuscript. For instance: Jabbour, J., Rahme, M., Mahfoud, Z. R., & El-Hajj Fuleihan, G. (2022). Effect of high dose vitamin D supplementation on indices of sarcopenia and obesity assessed by DXA among older adults: A randomized controlled trial. Endocrine, 76(1), 162–171. https://doi.org/10.1007/s12020-021-02951-3. I suggest for authors to recheck again the literature to include all studies. Response: The literature has been rechecked and references have been added. 5/ Discussion: I didn’t notice any paragraph that discuss the findings obtained in the review with other data. Authors should elaborate this paragraph to allow comparison between countries, ethnicities,.,, Response: The text has been rewritten and some discussion part has been added.

Reviewer 2 Report

In the manuscript nutrients-1823257 by Harahap et al, authors have presented a review discussing the role of Vitamin D and calcium in obesity and its comorbid conditions. Authors presented the literature mining strategy to highlight that only in vivo studies with scientific evidence for the correlation between calcium and vitamin D were included in this review. This review is comprehensive and shows authors efforts as over a hundred studies are cited. Overall, manuscript is well written. I have following point for authors to address.

1.    Line 11. “The effect of supplementation vitamin D and calcium…”. Please revise for readability.

2.    Line 17 “Adequate vitamin D 17 and calcium intake would be beneficial to safely control the global obesity epidemic.”. This line can be omitted as it is not informative. This message is already clear form the very first line of the abstract. Moreover, this is review article and this last line seems like a study conclusion.

3.    Figure 1. flow Chart. this flow is confusing. As it says 84 studies were included. What do author imply by this? This review would cite only 84 studies? I guess not. Or 84 studies were used for interpretation. Presented tables have more than 84 studies. In other words, what information are we getting with this flow chart and values over the information given in the Table 1.?  Please explain the purpose or else it can be removed.

4.    Line 79. “Subsequently…”. Subsequent of what?? Please consider revising.

5.    Line 84. “This suggests that the mechanism of action of vitamin D and calcium is interdependent”. Based on the description given in this paragraph, one would say calcium absorption is vitamin D dependent. It does not indicate "interdependent". Please consider explaining a bit.

6.    What is vitamin D-calcium axis? Please consider explaining this axis.

7.    Most of the subheadings in section 3 are backed up with 1 or 2 references. While these paragraphs a lot of information. Please consider providing references.

8.    Figure 2. Please consider elaborating the legend.

9.    Figure 2. The figure content does not clearly talk about the “interrelationship between vitamin D and Calcium".

10. Given this comprehensive review, I would recommend supplementing it with some mechanistic schematics. Specially about role of vitamin D in calcium uptake in different tissues.

Author Response

Dear Editor and Reviewers,

We are very grateful for your comments on our manuscript. We have revised the manuscript in accordance with your advice and carefully proofread it to minimize any errors.

Reviewer 2

1/   Line 11 “The effect of supplementation vitamin D and calcium…”. Please revise for readability.

Response: It has been rewritten.

2/  Line 17 “Adequate vitamin D 17 and calcium intake would be beneficial to safely control the global obesity epidemic.”. This line can be omitted as it is not informative. This message is already clear form the very first line of the abstract. Moreover, this is review article and this last line seems like a study conclusion.

Response: It has been rewritten.

3/ Figure 1. flow Chart. this flow is confusing. As it says 84 studies were included. What do author imply by this? This review would cite only 84 studies? I guess not. Or 84 studies were used for interpretation. Presented tables have more than 84 studies. In other words, what information are we getting with this flow chart and values over the information given in the Table 1.?  Please explain the purpose or else it can be removed.

Response: We used 84 studies for interpretation (now we have 86), these articles met the inclusion and exclusion criteria. Other articles were cited to describe the problem and for interpretation.

4/ Line 79 “Subsequently…”. Subsequent of what?? Please consider revising.

Response: It has been rewritten.

5/ Line 84 “This suggests that the mechanism of action of vitamin D and calcium is interdependent”. Based on the description given in this paragraph, one would say calcium absorption is vitamin D dependent. It does not indicate "interdependent". Please consider explaining a bit.

Response: It has been rewritten.

6/ What is vitamin D-calcium axis? Please consider explaining this axis.

Response: It has been explained in point 3.

7/ Most of the subheadings in section 3 are backed up with 1 or 2 references. While these paragraphs a lot of information. Please consider providing references.

Response: It has been rewritten.

 8/ Figure 2. Please consider elaborating the legend.

Response: Figure 2 has been changed and we believe that it is more clear.

  9/ Figure 2. The figure content does not clearly talk about the “interrelationship between vitamin D and Calcium".

Response: It has been changed.

10/ Given this comprehensive review, I would recommend supplementing it with some mechanistic schematics. Specially about role of vitamin D in calcium uptake in different tissues.

Response: We tried to show this relationship in Figure 2 and in the description.

Reviewer 3 Report

The concept of the review article is good, and it is a good topic for the audiences.

Please further elaborate uniqueness of this study by stating how your study fill the gap in this domain? Please indicated clearly the aim of the study in the introduction section.

What is the main motivation of the study?

The abstract and Conclusions sections must be re-consider.

Some sub-sections  of part 3. Physiological relationship between vitamin D and calcium, need to be further elaborate supported with relevant references only one reference per sub-sections is not relevant for such a review, e.g.

3.5. Vitamin Dcalcium axis in pancreas

3.6. Vitamin Dcalcium axis in liver

3.7. Vitamin Dcalcium axis in adipose tissue

Lines 182-185 - which data? there are no refrences to support the statement, and this sub-part must be further elaborate.

Lines 397-403 - references missing

Author Response

Dear Editor and Reviewers,

We are very grateful for your comments on our manuscript. We have revised the manuscript in accordance with your advice, and carefully proofread it to minimize any errors.

Reviewer 3

1/ Please further elaborate uniqueness of this study by stating how your study fill the gap in this domain? Please indicated clearly the aim of the study in the introduction section.

Response: It has been added.

2/ What is the main motivation of the study?

Response: The justification of this review has been added to the Introduction part.

3/ The abstract and Conclusions sections must be re-consider.

Response: These parts have been rewritten.

4/ Some sub-sections  of part 3. Physiological relationship between vitamin D and calcium, need to be further elaborate supported with relevant references only one reference per sub-sections is not relevant for such a review, e.g.3.5. Vitamin D–calcium axis in pancreas, 3.6. Vitamin D–calcium axis in liver, 3.7. Vitamin D–calcium axis in adipose tissue.

Response: This subsection has been rewritten.

5/ Lines 182-185 - which data? there are no refrences to support the statement, and this sub-part must be further elaborate.

Response: This part has been rewritten.

6/ Lines 397-403 - references missing

Response: It has been added.

Round 2

Reviewer 1 Report

All the comments were taken into consideration

The manuscript can be accepted now

Reviewer 3 Report

The authors adressed my recommandations.